# On the Expressive Power of Mixed-Curvature Representations in Product Manifolds

**Haitz Sáez de Ocáriz Borde**
University of Oxford
`chri6704@ox.ac.uk`

## Abstract

Mixed-curvature representations based on product manifolds of Euclidean, hyperbolic, and spherical spaces are widely used as latent geometries in representation learning and geometric deep learning. Their popularity stems from their analytical tractability and closed-form geodesic distances, which facilitate efficient backpropagation in deep learning. In the literature, they are often portrayed as richer representations than, for instance, hyperbolic embeddings. However, their expressive power is not well understood. In this work, we show that such mixed-curvature product manifolds impose rigid geometric constraints that fundamentally limit their expressive power. In particular, we prove that generic Riemannian manifolds cannot be locally represented isometrically by mixed-curvature product manifolds equipped with product metrics. The obstruction arises from curvature splitting: product metrics necessarily exhibit vanishing mixed sectional curvature (equivalently, flatness on mixed 2-planes), a rigidity that fails for a generic Riemannian metric.

## 1 Introduction

Product manifolds composed of Euclidean, hyperbolic, and spherical factors (often called constant-curvature model spaces) have gained popularity (Skopek et al., 2019; Gu et al., 2019; Giovanni et al., 2022; Borde et al., 2023a;c;b; Lu et al., 2023; Wang et al., 2021; Liu, 2025; Takeuchi et al., 2022) in representation learning and geometric deep learning (Bronstein et al., 2021; 2016; Borde & Bronstein, 2025). This is because they allow different components (or submanifolds) to encode distinct structural features while retaining closed-form geodesic distances essential for backpropagation in deep learning. Therefore, they are often presented as a richer alternative that generalizes constant curvature spaces such as hyperbolic (Nickel & Kiela, 2017; Ganea et al., 2018a;b; Tifrea et al., 2018; Sala et al., 2018; Liu et al., 2019; Shimizu et al., 2021; Kratsios et al., 2024) and spherical spaces (Esteves et al., 2020; 2023) which are widely used in neural network architectures and as non-Euclidean embedding spaces.

Despite their practical appeal and relatively widespread adoption, the limitations of such mixed-curvature product manifolds remain poorly understood in the geometric deep learning community. Hence, a natural question arises:

*Can mixed-curvature product manifolds represent arbitrary Riemannian geodesic distances?*

In this paper, we give a negative answer. We show that mixed-curvature product manifolds equipped with product metrics obey rigid curvature-splitting constraints: mixed 2-planes necessarily have zero sectional curvature. As a result, Riemannian metrics whose curvature is not product-splitting at a point $p$ cannot be locally represented isometrically by such products. The obstruction is purely local and independent of global topology, highlighting a fundamental limitation of commonly used product latent geometries.

## 2 Preliminaries: Riemannian Manifolds and Product Manifolds

We briefly recall the basic geometric notions needed throughout the paper and fix notation.

**Definition 1** (Smooth manifold). A *smooth manifold* of dimension $n$ is a Hausdorff, second countable topological space $M$ such that every point $p \in M$ admits a neighborhood homeomorphic to an open subset of $\mathbb{R}^n$, with smooth transition maps between overlapping charts ($C^\infty$).

**Definition 2** (Riemannian manifold). A *Riemannian manifold* is a pair $(M, g)$, where $M$ is a smooth manifold and $g_p : T_pM \times T_pM \to \mathbb{R}$ is a smoothly varying inner product on each tangent space $T_pM$, called the *Riemannian metric*.

**Definition 3** (Geodesic distance). Let $(M, g)$ be a Riemannian manifold. The *geodesic distance* (which is a metric in the metric geometry sense) between points $x, y \in M$ is defined by $d_M(x, y) = \inf_\gamma \int_0^1 \sqrt{g_{\gamma(t)}(\dot{\gamma}(t), \dot{\gamma}(t))}\, dt$, where the infimum is taken over all $\gamma : [0, 1] \to M$ satisfying $\gamma(0) = x$ and $\gamma(1) = y$.

**Definition 4** (Vector Fields). We denote by $\Gamma(TM)$ the set of all smooth *vector fields* on $M$, which are smooth mappings that assign a tangent vector $X_p \in T_pM$ to each point $p \in M$.

**Definition 5** (Levi-Civita Connection). The *Levi-Civita connection* is the unique map $\nabla : \Gamma(TM) \times \Gamma(TM) \to \Gamma(TM)$ that satisfies the conditions of being torsion-free and compatible with the metric $g$. For vector fields $X$ and $Y$, the quantity $\nabla_X Y$ is the covariant derivative of $Y$ in the direction $X$.

**Definition 6** (Riemann Curvature Tensor). The *Riemann curvature tensor $R$* is a $(1, 3)$-tensor field that measures the non-commutativity of the covariant derivative. For vector fields $X, Y, Z \in \Gamma(TM)$, it is defined as:

$$R(X, Y)Z = \nabla_X \nabla_Y Z - \nabla_Y \nabla_X Z - \nabla_{[X,Y]} Z, \tag{1}$$

where $[X, Y] = XY - YX$ is the Lie bracket of the vector fields.

**Definition 7** (Sectional Curvature). The *sectional curvature* $\sec_p(\Pi)$ is a measure of the curvature of a 2-dimensional plane $\Pi \subset T_pM$. If the plane $\Pi$ is spanned by the linearly independent vectors $\{u, v\}$, the sectional curvature is given by:

$$\sec_p(u, v) = \frac{g(R(u, v)v, u)}{g(u, u)g(v, v) - g(u, v)^2}. \tag{2}$$

This value is intrinsic to the plane and does not depend on the specific choice of basis $\{u, v\}$.

**Definition 8** (Cartesian product). Let $X_1, \ldots, X_K$ be sets. Their *Cartesian product* is the set $X_1 \times \cdots \times X_K = \{(x_1, \ldots, x_K) : x_k \in X_k \forall k\}$.

**Definition 9** (Product manifold). Let $(M_1, g_1), \ldots, (M_K, g_K)$ be Riemannian manifolds. Their product manifold is $M = M_1 \times \cdots \times M_K$ equipped with the product metric $g = g_1 \oplus \cdots \oplus g_K = \bigoplus_{k=1}^K g_k$. Geodesics and distances decompose across factors.

**Definition 10** (Mixed-curvature product manifold). A mixed-curvature product manifold based on constant curvature model spaces is a product of the form

$$\mathcal{P} = \mathbb{E} \times \left( \times_{i=1}^{n_{\mathbb{H}}} \mathbb{H}_i \right) \times \left( \times_{j=1}^{n_{\mathbb{S}}} \mathbb{S}_j \right), \tag{3}$$

where $\mathbb{E}$ is Euclidean space, each $\mathbb{H}_i$ is a hyperbolic space (with negative curvature), and each $\mathbb{S}_j$ is a spherical space (a hypersphere, with positive curvature), equipped with a product metric. We use a single Euclidean space with dimensionality $d_{\mathbb{E}}$ since taking the Cartesian product of Euclidean spaces simply results in another Euclidean space of higher dimensionality. Note $K = 1 + n_{\mathbb{H}} + n_{\mathbb{S}}$.

**Remark 1.** There exist multiple *models* (or realizations) of the same intrinsic Riemannian manifold. For instance, the Poincaré ball (or Poincaré disk in two dimensions), the Poincaré upper half-plane, and the hyperboloid models are all different models of hyperbolic space. These may all be used as part of the product manifold. Likewise, we can have stereographic projections of spherical space too.

**Definition 11** (Mixed-curvature distance). The squared (induced) geodesic distance on $\mathcal{P}$ satisfies $d_{\mathcal{P}}(x, y)^2 = \sum_{k=1}^K d_k(x_k, y_k)^2$, where $d_k$ are the factor geodesic distances and $x = (x_1, ..., x_K), y = (y_1, ..., y_K)$. This is the Pythagorean theorem generalized to product metric spaces.

**Remark 2** (Learning curvature via backpropagation). For computational tractability, one may normalize the hyperbolic and spherical factors to have unit negative and positive curvature, respectively, and learn curvature through scaling. In this case,

$$d_{\mathcal{P}}(x, y)^2 = \sum_{k=1}^K \alpha_k \hat{d}_k(x_k, y_k)^2 = \alpha_{\mathbb{E}} d_{\mathbb{E}}^2 + \sum_{i=1}^{n_{\mathbb{H}}} \alpha_{\mathbb{H}_i} \hat{d}_{\mathbb{H}_i}^2 + \sum_{j=1}^{n_{\mathbb{S}}} \alpha_{\mathbb{S}_j} \hat{d}_{\mathbb{S}_j}^2 \tag{4}$$

where $(\hat{\cdot})$ denotes quantities computed in the normalized spaces and $\alpha_k, \alpha_{\mathbb{H}_i}, \alpha_{\mathbb{S}_j} > 0$. This allows learning the (effective) curvature parameters via backpropagation through the distance computation alone, avoiding differentiation of the curvature through exponential/logarithmic maps (used in manifold optimization), which can be numerically delicate in practice. $\alpha_{\mathbb{E}}$ may be set to 1 but often a "relative scaling" for the Euclidean component is also learned.

**Remark 3** (Metric properties). $d_{\mathcal{P}}$ is a metric satisfying: (i) non-degeneracy: $d_{\mathcal{P}}(x, y) = 0$ if and only if $x = y$; (ii) symmetry: $d_{\mathcal{P}}(x, y) = d_{\mathcal{P}}(y, x)$; (iii) triangle inequality: $d_{\mathcal{P}}(x, y) \leq d_{\mathcal{P}}(x, z) + d_{\mathcal{P}}(z, y)$.

We formalize what it means for a product manifold to represent a Riemannian distance.

**Definition 12** (Local distance representation). A Riemannian manifold $(M, g)$ is said to be locally representable by a class of manifolds $\mathcal{C}$ if for every $p \in M$ there exists a neighborhood $U \ni p$ and $(N, h) \in \mathcal{C}$ such that $(U, d_g)$ is isometric to an open subset of $(N, d_h)$.

**Definition 13** (Isometry of metric spaces). Let $(X, d_X)$ and $(Y, d_Y)$ be metric spaces. A map $\varphi : X \to Y$ is called an *isometry* if it is a bijection onto its image and satisfies

$$d_Y(\varphi(x), \varphi(x')) = d_X(x, x') \quad \text{for all } x, x' \in X. \tag{5}$$

**Remark 4.** Here, the notion of isometry is purely metric and does not *a priori* require $\varphi$ to be smooth. However, when the metrics arise from Riemannian structures, any such local isometry between sufficiently small open sets is automatically smooth and preserves the Riemannian metric tensor.

## 3 RIGIDITY OF PRODUCT METRICS AND LOCAL NON-REPRESENTABILITY

We isolate a purely local obstruction to representing Riemannian distances by mixed-curvature product manifolds equipped with the *product metric*. The key point is that product metrics enforce curvature *splitting*: any 2-plane spanned by vectors from distinct factors has zero sectional curvature.

Let $(M, g)$ be a Riemannian product manifold as in Definition 9. Denote by $V_k \subset TM$ the subbundle tangent to the $k$-th factor, so that $TM = \bigoplus_{k=1}^{K} V_k$ orthogonally with respect to $g$. This implies that the metric $g$ has no "cross-terms" between different factors; specifically, $g(u, v) = 0$ whenever $u \in V_i$ and $v \in V_j$ for $i \neq j$. Or equivalently, the Riemannian product metric is block diagonal. Note that unlike in Definition 10, here $i$ and $j$ are used as generic indices to represent any two arbitrary factors, rather than to distinguish between types of manifolds.

**Lemma 1** (Product Levi–Civita connection). *Let $\nabla$ be the Levi–Civita connection of the product metric $g$ on $M = M_1 \times \cdots \times M_K$. We denote by $\Gamma(TM)$ the space of smooth vector fields on $M$, and by $\Gamma(V_k)$ the subbundle of vector fields tangent to the $k$-th factor. If $X \in \Gamma(V_i)$ and $Y \in \Gamma(V_j)$ are* lifted *(i.e. $X$ depends only on the $M_i$-coordinate and $Y$ depends only on the $M_j$-coordinate), then*

$$\nabla_X Y = \begin{cases} \nabla_X^{(i)} Y \in \Gamma(V_i), & i = j, \\ 0, & i \neq j, \end{cases} \tag{6}$$

*where $\nabla^{(i)}$ is the Levi–Civita connection of $(M_i, g_i)$.*

*Proof.* Recall that the Levi–Civita connection is uniquely characterized by the Koszul formula:

$$2g(\nabla_X Y, Z) = Xg(Y, Z) + Yg(X, Z) - Zg(X, Y) + g([X, Y], Z) - g([X, Z], Y) - g([Y, Z], X).$$

Assume $X \in \Gamma(V_i)$ and $Y \in \Gamma(V_j)$ are lifted with $i \neq j$. Since $g$ is a product metric, $V_i \perp V_j$ and thus $g(X, Y) = 0$ everywhere, hence $Zg(X, Y) = 0$ for any $Z$. Moreover, lifted vector fields from different factors commute, so $[X, Y] = 0$. For any test vector field $Z \in \Gamma(TM)$, decompose $Z = \sum_k Z_k$ with $Z_k \in \Gamma(V_k)$. If $k \neq i, j$ then $g(Y, Z_k) = g(X, Z_k) = 0$, so the first two terms vanish on $Z_k$, and the bracket terms also vanish by orthogonality. If $Z_i \in \Gamma(V_i)$ then $g(Y, Z_i) = 0$, and $[X, Z_i] \in \Gamma(V_i)$ is orthogonal to $Y$, so $g([X, Z_i], Y) = 0$; similarly for $Z_j \in \Gamma(V_j)$. Thus the right-hand side is zero for all $Z$, so $g(\nabla_X Y, Z) = 0$ for all $Z$, and by non-degeneracy $\nabla_X Y = 0$. The case $i = j$ follows by observing that for lifted fields tangent to the same factor, the Koszul formula reduces to that of $(M_i, g_i)$, hence $\nabla_X Y = \nabla_X^{(i)} Y \in \Gamma(V_i)$. $\square$

**Proposition 1** (Vanishing mixed sectional curvature). *Let $(M, g)$ be a Riemannian product with splitting $TM = \bigoplus_{k=1}^{K} V_k$. If $u \in (V_i)_p$ and $v \in (V_j)_p$ with $i \neq j$, then $\sec_p(\text{span}\{u, v\}) = 0$.*

*Proof.* Extend $u, v$ to local lifted fields $U \in \Gamma(V_i)$, $V \in \Gamma(V_j)$ with $[U, V] = 0$. By Lemma 1, $\nabla_U V = 0$ and $\nabla_V U = 0$, and $\nabla_V V \in \Gamma(V_j)$, hence again $\nabla_U(\nabla_V V) = 0$. Therefore $R(U, V)V = 0$ at $p$, i.e. $R(u, v)v = 0$, so the sectional curvature of $\text{span}\{u, v\}$ is zero. $\square$

To provide some visual context consider a cylinder ($\mathbb{S}^1 \times \mathbb{R}^1$), which is a product manifold. Notice how it has one curved direction (the circle) and one flat direction (the line). The sectional curvature of a plane spanning both directions is zero (you can unroll a cylinder into a flat sheet). Contrast this with a sphere ($\mathbb{S}^2$), which is not a product manifold, curves in all directions simultaneously, and cannot be unrolled without distortion.

We now apply this rigidity to mixed-curvature product manifolds as in Equation 3 equipped with the product metric $d_{\mathcal{P}}$ (with at least two nontrivial factors, so $K \geq 2$).

**Theorem 1** (Local non-representability). *Let $(M, g)$ be a smooth Riemannian manifold of dimension $n \geq 3$, and let $p \in M$. Assume there is* no *nontrivial orthogonal decomposition $T_p M = \bigoplus_{i=1}^{K} V_i$, $K \geq 2$, such that all mixed sectional curvatures vanish at $p$, i.e. $\sec_p(\text{span}\{u, v\}) = 0$ for all $u \in V_i$, $v \in V_j$, $i \neq j$. Then $(M, g)$ is not locally representable (as a metric space) by any product manifold endowed with its product Riemannian metric.*

*Proof.* Suppose $(U, d_g)$ is isometric (Definition 13) to an open set $(V, d_h)$ in a mixed-curvature product $(\mathcal{P}, h)$ via a distance-preserving bijection $\varphi : (U, d_g) \to (V, d_h)$ with $\varphi(p) = q$. By the local Myers–Steenrod theorem (Myers & Steenrod, 1939), $\varphi$ is smooth and satisfies $\varphi^* h = g$ on $U$; hence $d\varphi_p : T_p M \to T_q \mathcal{P}$ is a linear isometry and preserves sectional curvature of 2-planes. Because $(\mathcal{P}, h)$ is a nontrivial Riemannian product, $T_q \mathcal{P}$ splits orthogonally as $T_q \mathcal{P} = \bigoplus_{k=1}^{K} W_k$, and by Proposition 1 every plane spanned by $a \in W_i$ and $b \in W_j$ ($i \neq j$) has zero sectional curvature. Pull back the splitting by setting $V_i := d\varphi_p^{-1}(W_i)$. Then $T_p M = \bigoplus_k V_k$ orthogonally, and the mixed sectional curvatures vanish at $p$ by curvature preservation. Hence, contradicting the hypothesis. $\square$

**Remark 5** (What this rules out). The obstruction is local: any local isometry into a nontrivial product forces a tangent-space splitting with vanishing mixed sectional curvature at the basepoint. Thus, unless the geometry of $(M, g)$ is locally "product-like" (in this precise curvature sense), mixed-curvature product latent spaces cannot exactly reproduce its geodesic distances.

**Remark 6** (Curvature Sparsity in High Dimensions). In modern representation learning, the dimensionality of latent spaces is often large to accommodate high-dimensional feature vectors. However, Theorem 1 shows that for product manifolds equipped with the product metric, increasing the total dimensionality does not resolve the underlying representation problem. In particular, no amount of additional dimensionality can overcome the obstruction of Theorem 1 when the underlying geometry requires nonzero curvature on 2-planes that span multiple factors.

## 4 CONCLUSION

Our results demonstrate that mixed-curvature product manifolds equipped with the product metric impose rigid curvature-splitting constraints that prevent local isometric representation of Riemannian distances whose sectional curvature is not product-splitting at a point $p$, thereby limiting their expressive power. In short, we gain computational tractability and efficiency (closed-form distances) at the direct cost of expressivity: product metrics cannot model non-separable curvature interactions across factors. While such manifolds remain useful when the data geometry is approximately separable, they are not locally universal in the isometric sense for Riemannian distances. We leave quantitative approximation guarantees (e.g., lower bounds on bi-Lipschitz distortion) to future work.

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
