# OpenReview forum: "On the Expressive Power of Mixed-Curvature Representations in Product Manifolds"
_ICLR.cc/2026/Workshop/GRaM — ICLR 2026 Workshop GRaM Poster_

### Official Review · Reviewer_qWWk · 2026-02-09
**Good theoretical paper**

**Rating:** 7
**Confidence:** 3

**Review:**

This paper demonstrates that mixed-curvature product manifolds equipped with the product metric impose rigid curvature-splitting constraints that prevent local isometric representation of Riemannian distances, where sectional curvature is not product-splitting at a point $p \in M$, which limits their expressive power.
The authors establish the following trade-off: computational tractability and efficiency are gained at the direct cost of expressivity; product metrics cannot model non-separable curvature interactions across factors.
The conclusion is that, despite being useful for cases with approximately separable data geometry, mixed-curvature product manifolds are not locally universal in the isometric sense for Riemannian distances.

Strengths: clarity, presentation and organisation of the paper, as well as its relevance to the GRaM themes. The analysis and conclusions appear to be original, given the clearly provided definitions and scope.

Weaknesses: given the scope and future work remark, no substantial weaknesses were identified.

**Pmlr Suitability:**

NA

---

### Official Review · Reviewer_HMpn · 2026-02-15
**Correct and wellwritten paper, but importance of results unclear**

**Rating:** 5
**Confidence:** 3

**Review:**

The article is concerned with mixed curvature product manifolds, which are becoming more and more popular for representation learning. The authors pose and answer a 'universality' question: Can such manifolds locally represent any Riemannian manifold, in a 'geodesic' sense. The argument is clean, but also relatively straightforward: The authors show that the sectional curvature must vanish along any plane spanned up by vectors in distinct factors of the tangent space (corresponding to the factorization of the product manifold).

Strengths: The exposition is well-written, and the proofs are correct. The topic, in particular the ambition to understand fundamental limitations of widely used model, is relevant to the GRaM community.

Weaknesses: The impact and significance are unclear to me. The results are without the interpretation in the context of product manifolds being used for machine learning models quite simple. This is of course not disqualifying, but since the authors do not discuss when their condition actually disqualifies a manifold at all. The paper would benifit from giving one concrete example of a geometry that cannot be captured - a more general description of such geometry would be even better.

**Pmlr Suitability:**

NA

---

### Meta-Review · Area_Chair_6Jxh · 2026-02-26

**Decision:**

Accept

**Metareview:**

This paper shows that mixed-curvature product manifolds are limited in their expressive power, since the sectional curvature vanishes on mixed 2-planes. It's a clean result that gives novel insights on mixed-curvature embeddings. We are happy to accept it for our tiny paper track!

**Relevance To Proceedings:**

Tiny paper — does not apply

**Relevance To Workshop:**

Yes — suitable for GRaM

---

### Decision · Program_Chairs · 2026-03-02

Accept (Poster)